# Clinical Utility of Second-Look FDG PET-CT to Stratify Incomplete Metabolic Response Post (Chemo) Radiotherapy in Oropharyngeal Squamous Cell Carcinoma

**DOI:** 10.3390/cancers15020464

**Published:** 2023-01-11

**Authors:** Sarah Billingsley, Zsuzsanna Iyizoba, Russell Frood, Sriram Vaidyanathan, Robin Prestwich, Andrew Scarsbrook

**Affiliations:** 1Department of Radiology, Leeds Teaching Hospitals NHS Trust, Leeds LS9 7TF, UK; 2Leeds Institute of Health Research, Faculty of Medicine & Health, University of Leeds, Leeds LS9 7TF, UK; 3Department of Clinical Oncology, Leeds Cancer Centre, Leeds LS9 7TF, UK

**Keywords:** PET-CT, head and neck cancer, radiotherapy, chemotherapy, recurrence

## Abstract

**Simple Summary:**

Incomplete imaging response following non-surgical treatment for head and neck cancer is common, and optimal management is uncertain. This single-centre study assessed the value of performing a second PET-CT scan a few months later in patients with uncertain findings initially after treatment and showed that in most cases, the changes resolved or stayed the same and were not due to residual cancer. This approach could be used to spare unnecessary surgery when there is initial uncertainty.

**Abstract:**

Background: Incomplete response on FDG PET-CT following (chemo)radiotherapy (CRT) for head and neck squamous cell carcinoma (HNSCC) hinders optimal management. The study assessed the utility of an interval (second look) PET-CT. Methods: Patients with oropharyngeal squamous cell carcinoma cancer (OPSCC) treated with CRT at a single centre between 2013 and 2020 who underwent baseline, response, and second-look PET-CT were included. Endpoints were conversion rate to complete metabolic response (CMR) and test characteristics of second-look PET-CT. Results: In total, 714 patients with OPSCC underwent PET-CT post-radiotherapy. In total, 88 patients with incomplete response underwent second-look PET-CT a median of 13 weeks (interquartile range 10–15 weeks) after the initial response assessment. In total, 27/88 (31%) second-look PET-CTs showed conversion to CMR, primary tumour CMR in 20/60 (30%), and nodal CMR in 13/37 (35%). In total, 1/34 (3%) with stable tumour/nodal uptake at the second-look PET-CT relapsed. Sensitivity, specificity, positive (PPV), and negative predictive value (NPV) of second-look PET-CT were 95%, 49%, 50%, and 95% for tumour and 92%, 50%, 50%, and 92% for nodes, respectively. Primary tumour progression following CMR occurred in one patient, two patients with residual nodal uptake at second-look PET-CT progressed locoregionally, and one patient developed metastatic disease following CMR in residual nodes. Conclusion: Most patients undergoing second-look PET-CT converted to CMR or demonstrated stable PET signal. NPV was high, suggesting the potential to avoid unnecessary surgical intervention.

## 1. Introduction

Oropharyngeal cancer (OPSCC) is becoming increasingly prevalent worldwide due to the rising incidence of high-risk human papilloma virus. OPSCC led to 48,100 deaths in 2020 alone [1]. For most patients with OPSCC, curative-intent treatment is primarily with (chemo)radiotherapy. Treatment response assessment is important in non-surgically treated patients to accurately identify patients who may require additional salvage surgery from those who can be managed conservatively with clinical follow-up. Residual masses are commonly demonstrated on anatomical imaging (CT and MRI) and may be secondary to post-treatment changes rather than residual disease [2,3].

2-[Fluorine-18]-fluoro-2-deoxy-D-glucose (FDG) positron emission tomography-computed tomography (PET-CT) combines anatomical imaging with functional information and is increasingly used for response assessment in a range of cancers because the superior accuracy of the technique compared to conventional imaging [4,5,6]. A prior large randomised controlled trial in the UK demonstrated that follow-up guided by PET-CT surveillance had a non-inferior survival rate compared to planned neck dissection and was more cost-effective [3]. FDG PET-CT is now widely employed as a standard of care in the assessment of HNSCC patients’ post-treatment and is advocated in international imaging and clinical practice guidelines [3,7].

FDG PET-CT can be useful in the assessment of residual neck masses since those which are metabolically inactive are unlikely to represent residual disease. Multiple centres have employed FDG PET-CT as a tool for post-treatment management and identification of patients requiring surgical follow-up [8,9,10,11,12,13]. Initial post-treatment PET-CT examinations can have a limited positive predictive value (PPV) due to metabolic activity related to post-radiotherapy inflammation [2]. This can increase anxiety over the presence of residual disease, potentially leading to unrequired intervention and the optimal management strategy in this scenario remains uncertain. Currently, there is a lack of consensus on how to reliably stratify these patients. Recent studies have shown that performing a second-look post-treatment PET-CT in patients with initially equivocal responses, either in the primary tumour or nodal disease, can identify those who go on to complete metabolic response and therefore benefit from clinical follow-up instead of progression to surgical exploration [11,14].

The increased time interval between the completion of initial treatment and subsequent interim PET-CT allows for post-treatment changes and reactive FDG uptake to reduce, improving the negative predictive value and specificity when compared to the initial post-treatment PET-CT [2]. Our centre previously reported the feasibility of a second-look PET-CT in an initial cohort of patients with incomplete response to post-treatment PET-CT with promising results [15]. The aim of this study was to further assess the clinical utility of a second-look interval PET-CT in an expanded cohort of patients treated between 2013 and 2020.

## 2. Materials and Methods

### 2.1. Ethics Approval

All patients included in the analysis provided consent at the time of imaging for their anonymised FDG PET-CT imaging data to be used in research and service development projects. Formal Ethics Committee approval was obtained for use of radiological imaging and structured clinical data for all cancer patients treated at our institution (RCD-Onc: Enhancing understanding and prediction of cancer outcomes with baseline characteristics from routinely collected data, Integrated Research Application Approval Number 277122).

### 2.2. Patient Selection

Consecutive adult patients with histologically proven OPSCC who were treated with (chemo)radiotherapy between July 2013 and May 2020 at Leeds Cancer Centre and underwent (i) baseline (pre-treatment), (ii) response assessment, and (iii) second-look PET-CTs following incomplete response (positive or equivocal scan) were included. Patients were identified from an institutional database. The decision for a second-look PET-CT was made on a patient-by-patient basis at the discretion of a locoregional multidisciplinary team meeting (MDT) based on clinical risk, endoscopic examination, and radiological findings.

### 2.3. Staging

The staging was completed via a combination of nasoendoscopy, examination under anaesthetic with biopsy (if required), multiparametric MRI neck, FDG PET-CT, and, in some patients, an additional contrast-enhanced CT neck and thorax. Over the study period, American Joint Committee on Cancer TNM staging was updated from version 7 to 8. To maintain data collection cohesion, TNM 7 was employed throughout.

### 2.4. (Chemo)Radiotherapy

Radiotherapy techniques and the delineation of target areas changed over the course of the study period. During the initial period, a compartmental approach to target delineation was adopted [16]. Since 2016, a volumetric approach to outlining was routinely utilised, based upon primary tumour and involved lymph nodes +10 mm to high dose clinical target volume (CTV) and lymph node levels within elective dose CTVs. In 2018, a ‘5 + 5′ geometric approach to target delineation was adopted in line with international consensus guidelines [17]. The planning target volume (PTV) was created by auto-expansion of the CTV by 4 mm. Institutional protocols were followed with a radical treatment dose of 70 Gy in 35 fractions over 7 weeks (for concurrent chemoradiotherapy) or 65 Gy in 30 fractions over 6 weeks (for radical radiotherapy without chemotherapy), with lower doses to prophylactic dose regions (54–63 Gy in 30–35 fractions over 6–7 weeks) in line with Royal College of Radiologists guidelines [18]. Treatment was delivered with a volumetric arc therapy (VMAT) technique.

Patients under 70 years of age with no contraindication to platinum-based chemotherapy received 100 mg/m^2^ every three weeks as per local protocol. This regimen was administered alongside the aforementioned radiotherapy treatment (Table 1).

### 2.5. Response Assessment

Response evaluation was routinely assessed 4 months after completion of (chemo)radiotherapy and involved clinical examination, nasendoscopy (if required), and an initial post-treatment FDG PET-CT [4]. PET-CT results were discussed at the locoregional MDT when an incomplete response (as defined in *Categorisation of FDG PET-CT response* section below) was demonstrated. Most patients with probable residual disease on response assessment PET-CT were not considered for a second-look PET-CT and instead worked up with biopsy and cross-sectional imaging to evaluate the possibility of surgical salvage. A few patients who had suspected residual disease not considered surgically salvageable based on their disease extent at baseline, co-morbidities, or their individual wishes were referred for second-look PET-CT as a simple way of determining disease progression. Most patients who underwent second-look PET-CT had more equivocal/indeterminate findings at the initial response assessment.

### 2.6. Imaging Protocol

FDG PET-CT scans were performed as part of routine clinical practice with coverage from the skull vertex to the upper thigh in all patients. Imaging was acquired on three scanners during the study period, including a 64-slice Philips Gemini TF64 scanner (Philips Healthcare, Best, Netherlands), a 64-slice Discovery 690 scanner (GE Healthcare, Chicago, IL, USA), or a 64-slice Discovery 710 scanner (GE Healthcare, Chicago, IL, USA). Patients were fasted for 6 h before administration of Fluorine-18 FDG intravenously (4 MBq/kg). Serum blood glucose levels were measured prior to radiotracer administration, and if >10 mmol/L, the examination was not performed. PET was performed 60 min after the radiotracer injection. Following the injection, the patient underwent silence protocol to minimise physiological head and neck tracer uptake for the duration of the scan. Standard proprietary time-of-flight iterative reconstruction algorithms were used for PET imaging data. The CT component was performed using a low-dose unenhanced protocol with the following settings: 140 Kv, 80 mAs, tube rotation 0.52 per s, pitch 6, section thickness 3.75 mm.

### 2.7. Categorisation of FDG PET-CT Response

Clinical reports of PET-CT scans were primarily used to collate primary tumour and nodal disease response assessment categories. All FDG-PET CT scans were reported by a team of experienced radiology consultants with a minimum of 5 years of experience in interpreting oncological PET-CT studies. Standard-of-care imaging evaluation included semi-quantitative analysis of the maximum standardised uptake value (SUV_max_) of both primary tumour and nodal disease sites, as well as any potential distant disease sites along with a comparison to background physiological tracer activity within the liver and mediastinal blood pool (MBP) for reference [19]. PET-CT imaging was independently reviewed in any case where insufficient details were included in the clinical report by a dual-certified Radiologist and Nuclear Medicine Physician with more than 15 years of experience. Post-treatment FDG PET-CT studies were further sub-categorised as ‘positive’, ‘equivocal’, or ‘negative’ at both the primary tumour and nodal sites. A ‘positive’ result was recorded in the event of focal uptake greater than background physiological liver uptake. An ‘equivocal’ classification was assigned to cases with reduced tracer uptake from baseline imaging with residual activity above MBP but below liver activity. A ‘negative’ result was recorded in the event of no FDG uptake or FDG uptake below MBP. The same categorisation techniques were employed to characterise second-look FDG PET-CT studies (Figure 1).

### 2.8. Analysis and Statistics

The primary endpoints were the conversion rate to a ‘negative’ scan and test characteristics of the second-look PET-CT. Follow-up duration was defined from the day of the final fraction of radiotherapy treatment. Final disease status was determined from institutional electronic patient records, records of clinical assessment, and radiological and histological reports where biopsy indicated. 2 × 2 tables were created with clinicopathological data input to generate sensitivity, specificity, negative predictive value (NPV), and PPV.

## 3. Results

Patient demographics are listed in Table 1.

### Outcomes

In total, 714 patients with OPSCC cancer were treated with (chemo)radiotherapy within the study period. In total, 88 (12.3%) of these patients underwent a second-look PET-CT. In total, 53/88 were p16 positive, 15/88 were p16 negative, and 20/88 did not have a recorded HPV status. The median baseline primary tumour SUV_max_ was 13.1 (range of 4.4–27), and the mean baseline nodal SUV_max_ was 8.5 (range of 0–18.5). Median time from the end of treatment to the initial response assessment PET-CT was 17 weeks (interquartile range (IQR) of 16–17.5 weeks). In total, 44/88 (50%) patients demonstrated residual FDG activity at the primary tumour site alone, 21/88 (24%) demonstrated residual nodal FDG uptake, 16/88 (18%) demonstrated both primary and nodal uptake, 5/88 (6%) demonstrated FDG avidity in clinically and radiologically inconclusive lung nodules (Table 2). The median SUV_max_ of the residual primary tumour uptake was 4.5 (range of 2.2–10.7). The median SUV_max_ of residual nodal sites was 3 (range of 1.6–12.5). Median follow-up from the time of the last radiotherapy fraction to the date of the last formal follow-up or date last known to be alive for the second-look cohort was 23 months (IQR 14–33 months).

The median time from primary response assessment PET-CT to second-look imaging was 13 weeks (IQR 10–15 weeks). Overall, 27/88 ‘second-look’ scans demonstrated complete metabolic response (CMR, ‘negative’ categorisation). Conversion to CMR at second look in patients with residual FDG uptake at the primary tumour site occurred in 20/60 (33%). In total, 1/20 (5%) of these patients had biopsy-confirmed primary tumour relapse 16 months later, declined salvage surgery, and subsequently died (Figure 2). In total, 20/60 (33%) patients with primary tumour site uptake demonstrated stable FDG activity between the first and second-look PET-CTs, and none of the patients subsequently had disease relapse during the follow-up period. Another 20 patients (33%) with residual primary site uptake at initial response assessment PET-CT had local disease progression on second-look PET-CT.

In total, 13 of 37 (35%) patients with residual nodal uptake converted to CMR at second-look PET-CT, of which 2 subsequently developed distant disease relapse. In total, 14/37 (38%) demonstrated stable FDG uptake on second-look PET-CT (Table 3).

In total, 1 of these (7%) patients had a subsequent nodal relapse and underwent a salvage neck dissection but later died. In total, 2/33 (6%) patients with CMR at second-look PET-CT later developed distant disease progression; one underwent salvage resection and radiotherapy for lung metastases and remained disease free at last clinical contact, the other developed multifocal intra-thoracic disease 1 year after the second-look PET- CT and subsequently died.

The sensitivity, specificity, PPV, and NPV, respectively, of second-look PET-CT in primary tumour CMR were 95%, 49%, 50%, and 95%, and for nodal CMR, they were 92%, 50%, 50%, and 92%, respectively.

Lung metastases were confirmed in 5/5 (100%) patients on the second-look PET-CT when the concern on the primary post-treatment PET-CT were inconclusive, faintly FDG-positive lung nodules.

Of the 626 patients (88%) who did not undergo a second-look PET-CT median follow-up was 31 months (IQR 33–76 months). In total, 470 patients (75%) with CMR at initial response assessment PET-CT remained disease free; 155 patients (25%) had residual disease or subsequent disease progression in the primary tumour (11 patients), regional nodes (76 patients) or distant metastatic disease (75 patients). Successful salvage surgery was performed in 10 patients with neck dissection (9 patients) or lung wedge resection (1 patient). Two further patients had successful salvage radiation treatment to the contralateral neck or solitary distant metastasis. In addition, six patients had salvage treatment but subsequently progressed. One patient died of cardiac arrest shortly after treatment.

## 4. Discussion

This study demonstrates the application of a second-look FDG PET-CT approach can robustly and safely guide follow-up in patients with indeterminate residual uptake at primary or nodal disease sites on initial post-treatment imaging. The PPV was low at 50% for both primary and nodal CMR. This is likely multifactorial due to residual post-radiation inflammation and evolving real-world practice on categorisation of residual uptake during the study period with referrals for second-look PET-CT not rigorously stratified by degree of FDG activity into equivocal (below liver uptake) with a very low recurrence rate and more intense uptake with a higher incidence of residual disease. The NPV, however, is high, suggesting that a second-look PET can be safely utilised to stratify patients into appropriate management pathways and prevent overtreatment. This is concordant with the existing literature on response assessment for HNSCC [4,9,20,21]. The current study reaffirms that the deployment of functional imaging optimally guides management without the need for surgical re-exploration in the form of planned neck dissection in cases of complete metabolic response [11,22].

Uncertainty remains in the management of patients who demonstrate residual activity within either the primary or lymph node site on initial post-treatment PET-CT. There is also a lack of consensus as to the optimal timing of the PET-CT [2,22]. A scan performed too early after CRT increases the false positive rate, given the unresolved inflammatory FDG uptake, while a late scan runs the risk of delaying treatment of residual/recurrent disease. Studies have shown that scans performed less than 7 weeks post-treatment are less likely to give accurate results than those performed later [23], while a meta-analysis demonstrated a higher sensitivity for PET-CT performed over 12 weeks post-treatment [24]. Based on these data, 12 weeks is the target for the performance of the initial post-treatment PET-CT. Even if post-treatment PET-CT is performed at a slightly deferred time point (16 weeks), as is the routine practice at our centre [4], some patients still demonstrate equivocal/indeterminate activity at the primary tumour site, lymph nodes, or alternative areas raising concern for metastases. Prior publications have reported equivocal nodal uptake in 11–14% of cases, even on a 16-week response assessment PET-CT [4,25].

Importantly in this study, the NPV for second-look PET-CT was very high (100%) for primary tumour sites and 92% at nodal sites. The data suggest that many patients with equivocal results on the first post-treatment PET-CT will go on to convert to CMR, negating the need for follow-up neck dissection or invasive procedures such as EUA.

In total, 30% of our patients demonstrated a primary site conversion to CMR, and 35% of patients had nodal site CMR on the second-look PET-CT. The conversion rate was lower than previously reported by our centre (56% and 74% for the primary site and nodal site, respectively) [15]. These metrics are also lower compared to other groups. Vainshtein et al. [21] reported CMR of 95% and 100% for primary and nodal sites, respectively, on surveillance PET-CT performed approximately 3 months following initial response assessment imaging, although a third of their patient cohort had near CMR on the initial response assessment PET-CT. In another study, Iovoli et al. [26] performed repeat PET-CT a median of 91 days post initial incomplete response on PET-CT post CRT in 57 patients with head and neck cancer (47% had OPSCC), 26 patients (48%) converted to CMR with none subsequently relapsing. Liu et al. [14] reported a 71% conversion to CMR in 41 patients with OPSCC and inconclusive nodal response after a repeat PET-CT at 16 weeks. This suggests that the optimal timepoint for second-look imaging might be slightly later than 3 months. This study included patients with both HPV-positive and HPV-negative OPSCC. Previous studies investigating the clinical utility of second-look PET-CT were concerned primarily with HPV-positive head and neck cancer, and therefore these results may not be directly comparable [27]. Recent studies have investigated HPV-negative cohorts, and the initial data suggest that PET-CT is useful in monitoring treatment response [25]. With this caveat, the current study demonstrated similar numbers of patients with recurrence after having incomplete or stable responses.

One patient (1/20) demonstrating CMR at the primary site on second-look PET-CT developed local recurrence 16 months later. This patient declined salvage neck surgery and subsequently died. Two patients with a complete nodal response on the first response PET went on to later develop distant disease progression. One of these underwent a successful salvage procedure and radiotherapy for lung metastases, remaining disease-free at the time of writing, with a most recent CT thorax completed in January 2022 demonstrating no recurrence. The other developed multifocal intra-thoracic disease one year after the second-look PET-CT and died 7 days after diagnosis of lung metastases. In total, 1/14 patient with equivocal nodal response who was also equivocal on second-look PET relapsed and underwent a salvage neck dissection, ultimately progressing post-surgery and subsequently dying.

The study has some limitations similar to other papers based on real-world data. It is a retrospective, single-centre evaluation with no pre-determined standardised structured approach in selecting patients for a second-look PET-CT. However, in accordance with best clinical practice, all patients were selected based on the opinion of experienced PET reporters and discussion at a specialist MDT meeting. There were also cases of FDG-avid lung nodules and subsequent uncertainty over whether these were metastases or unrelated findings, which was not included as an indication in prior work. Not all patients underwent p16 expression testing, and a small number of patients (4%) in the study cohort had atypical oropharyngeal primary sites. We acknowledge that there is currently no clear consensus on the best methodology for HPV testing [28]. There is also a lack of consensus on the best method for evaluating PET-CT response post CRT in HNSCC, and a variety of different interpretative criteria have been proposed, but we have previously demonstrated that they have comparable accuracy, and none have been shown to be consistently superior [19]. A semi-quantitative approach was used in the study in order to reduce the impact of subjectivity on reporting, but some inherent variability based on reporter opinion will remain.

## 5. Conclusions

This study suggests that the majority of OPSCC patients who have an incomplete response on an initial post-treatment PET-CT have either CMR or stable uptake on interval re-assessment and do not subsequently relapse. This approach avoids the need to perform unnecessary salvage surgery on patients without residual disease and could be important in streamlining patient management as well as reducing morbidity, but it requires further prospective evaluation.

## Figures and Tables

**Figure 1 cancers-15-00464-f001:**
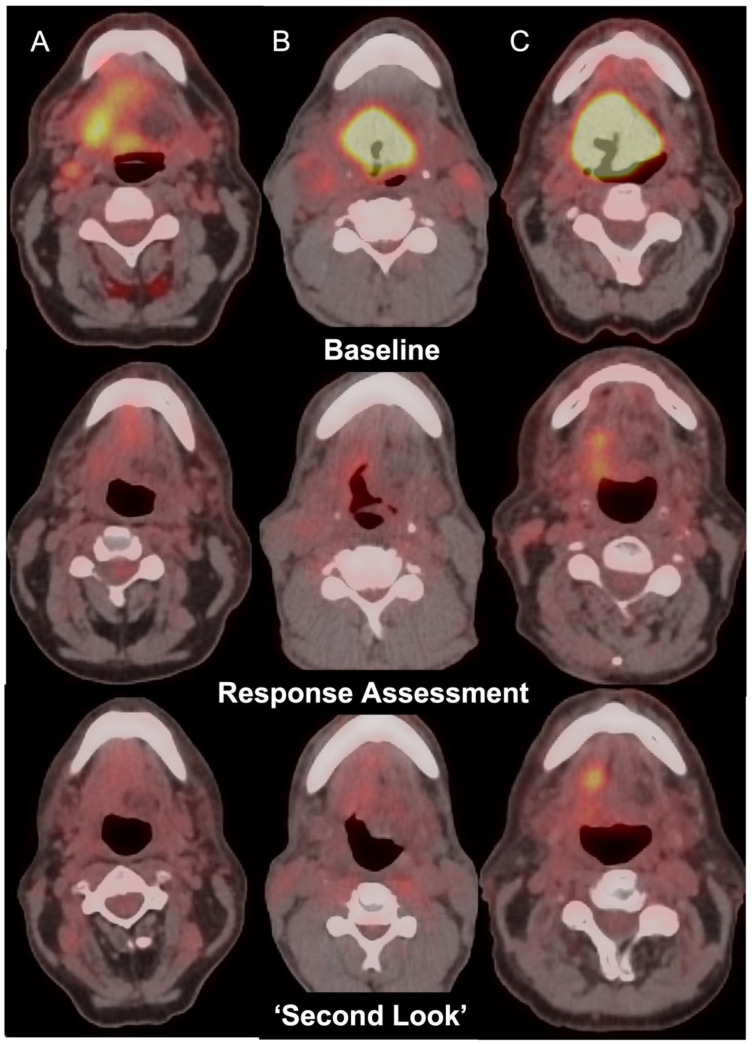
Axial fused FDG PET-CT images in 3 patients with oropharyngeal carcinoma (Top row, Columns A, B, C) with incomplete response at the primary tumour site on post-treatment FDG PET-CT (middle row) showing negative (**A**), equivocal (**B**), or positive (**C**) changes on ‘second-look’ imaging (bottom row).

**Figure 2 cancers-15-00464-f002:**
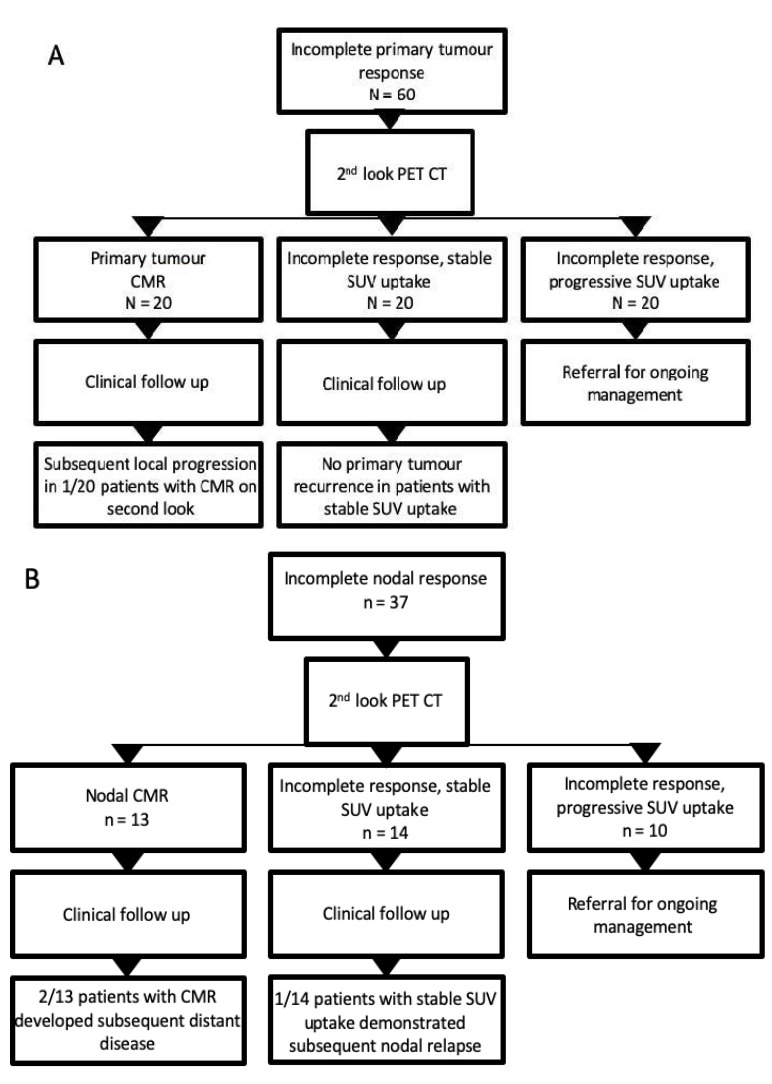
Flow chart illustrating outcomes of patients with incomplete primary tumour or nodal response on initial response assessment PET-CT. (**A**) Outcome of patients with residual FDG uptake at the primary tumour site on initial post-treatment PET CT; (**B**) outcome of patients with residual nodal uptake at initial response assessment PET-CT.

**Table 1 cancers-15-00464-t001:** Patient cohort characteristics.

	*n* = 88	No.
Age	Mean	61
	Range	37–78
Gender	Female	22 (25%)
	Male	66 (75%)
Smoking	Current smoker	30 (34%)
	Ex-smoker	26 (30%)
	Non-smoker	11 (12%)
	Not recorded	21 (24%)
Tumour site	Tonsil	52 (59%)
	Base of tongue	32 (36%)
	Soft palate	2 (2%)
	Vallecula	1 (1%)
	Posterior pharyngeal wall	1 (1%)
T stage	T1	8 (9%)
	T2	31 (35%)
	T3	13 (15%)
	T4	36 (41%)
N stage	N0	11 (12%)
	N1	9 (10%)
	N2a	7 (8%)
	N2b	33 (38%)
	N2c	26 (30%)
	N3	2 (2%)
HPV status	Positive	53 (60%)
	Negative	15 (17%)
	Unknown	20 (23%)
Radiotherapy	Radiotherapy alone	21 (24%)
	Chemoradiotherapy	67 (76%)

**Table 2 cancers-15-00464-t002:** Distribution of residual uptake on initial response assessment FDG PET-CT.

FDG Avidity on Initial Post-Treatment PET-CT	
Site	Patient Number (n)
Primary site only	44
Lymph node only	21
Primary site and lymph nodes	16
Lung nodules	5
Other	2

**Table 3 cancers-15-00464-t003:** Comparison of response categories between initial and second-look PET-CTs.

Second-Look PET Activity		
	Active Primary on Initial Response Assessment PET-CT	Active Lymph Node on Initial Response Assessment PET-CT
**Progression**	20	10
**Stable**	20	14
**Complete response**	20	13

## Data Availability

The data presented in this study are available on request from the corresponding author. The data are not publicly available due to institutional data-sharing restrictions.

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
