# Peer review of "Clinical Utility of Second-Look FDG PET-CT to Stratify Incomplete Metabolic Response Post (Chemo) Radiotherapy in Oropharyngeal Squamous Cell Carcinoma"

_cancers, 2023, doi:10.3390/cancers15020464_

Round 1
Reviewer 1 Report
In this manuscript the authors seek to build upon their previous work examining the utility of a second-look PET-CT for H&N cancer after an initial incomplete metabolic response, now with a cohort of patients over double the size of their former publication. While this study does not provide novel findings, the topic under investigation is still unclear in the oncology community and only sparse literature exists on this topic. This manuscript provides a welcome addition to an important topic in the post-treatment management of H&N cancer.
A few minor points below:
1. Consider adding percentages for each category in Table 1.
2. Consider adding median follow up time (with a range) for the total cohort since the results rely upon enough follow up time to accrue for disease failure events to happen.
3. Consider stratifying the predictive values and/or conversion to CMR by HPV-status if the number of patients in each group allows for it. If the findings are interesting could expand upon this in the discussion.
4. Consider adding this manuscript to discussion that similarly examined both HPV positive and negative patients in this context (https://doi.org/10.3390/cancers13061461)
Author Response
Reviewer 1
In this manuscript the authors seek to build upon their previous work examining the utility of a second-look PET-CT for H&N cancer after an initial incomplete metabolic response, now with a cohort of patients over double the size of their former publication. While this study does not provide novel findings, the topic under investigation is still unclear in the oncology community and only sparse literature exists on this topic. This manuscript provides a welcome addition to an important topic in the post-treatment management of H&N cancer.
Thank you for these helpful comments.
A few minor points below:
- Consider adding percentages for each category in Table 1.
Percentages have been added to Table 1 as suggested.
- Consider adding median follow up time (with a range) for the total cohort since the results rely upon enough follow up time to accrue for disease failure events to happen.
These have been added to the Outcomes section of the results as suggested for both the second look cohort and the whole study cohort. Outcome data for the 626 patients who did not have second look PET-CT have also been added for transparency/completeness.
- Consider stratifying the predictive values and/or conversion to CMR by HPV-status if the number of patients in each group allows for it. If the findings are interesting could expand upon this in the discussion.
Thanks for this suggestion. As this is quite a small cohort with only 15 (17%) patients with HPV negative disease this aspect can’t really be accurately assessed. We have performed Chi-squared analysis comparing complete metabolic response on second look PET-CT between HPV positive and HPV negative patients in the study dataset and there was no statistically significant difference between groups (p = 0.92). Given the small sample size and negative findings we do not feel this substantially adds anything to the manuscript and as a result have not added this to the results.
- Consider adding this manuscript to discussion that similarly examined both HPV positive and negative patients in this context (https://doi.org/10.3390/cancers13061461)
Thanks for this suggestion. We have amended the relevant section of the discussion and added this reference.
Reviewer 2 Report
Thank you for the opportunity to review your manuscript. The submitted manuscript works to address a relevant topic and issue among physicians treating oropharynx cancer, namely what to do when a post-treatment PET-CT demonstrates equivocal results. Specifically, this study looks at the metabolic response outcome when obtaining a second PET-CT a few months after the first. This is a retrospective study which follows patients who underwent a “second look” PET-CT after incomplete metabolic response was noted on the first post-treatment scan. The authors demonstrated that ~1/3 of patients who were re-scanned demonstrated complete metabolic response, which was similar between both the primary site and cervical lymph nodes. Of the 88 patients studied only 4 progressed with either locoregional or metastatic disease. The authors conclude that the negative predictive value of a repeat scan is high and could help avoid unnecessary procedures.
I do not have any major criticisms of this work. I think it is logical and valuable, and the authors present a sound argument with the number of available patients they have to study. There is a lot of uncertainty that surrounds how to survey and manage patients with PET-positive disease after cancer treatment, and this study helps describe one possible option, which is repeating a PET scan at a second interval. I will consider the results of this study further with patients in my own practice.
Author Response
Reviewer 2
Thank you for the opportunity to review your manuscript. The submitted manuscript works to address a relevant topic and issue among physicians treating oropharynx cancer, namely what to do when a post-treatment PET-CT demonstrates equivocal results. Specifically, this study looks at the metabolic response outcome when obtaining a second PET-CT a few months after the first. This is a retrospective study which follows patients who underwent a “second look” PET-CT after incomplete metabolic response was noted on the first post-treatment scan. The authors demonstrated that ~1/3 of patients who were re-scanned demonstrated complete metabolic response, which was similar between both the primary site and cervical lymph nodes. Of the 88 patients studied only 4 progressed with either locoregional or metastatic disease. The authors conclude that the negative predictive value of a repeat scan is high and could help avoid unnecessary procedures.
I do not have any major criticisms of this work. I think it is logical and valuable, and the authors present a sound argument with the number of available patients they have to study. There is a lot of uncertainty that surrounds how to survey and manage patients with PET-positive disease after cancer treatment, and this study helps describe one possible option, which is repeating a PET scan at a second interval. I will consider the results of this study further with patients in my own practice.
Many thanks for your helpful and positive comments.
Reviewer 3 Report
Patient selection:
Data on the 714-88= 626 patients who did not go to a second-look PET are missing. Are all those CMR or are some patients having progressive disease and thus subject to salvage treatment? It should be noted if all patients with positive or equivocal scans at 16 weeks were subject to second-look, and if they were it shoud be discussed what the purpose of the first scan at 16 weeks is.
Response assesment:
Response evaluation at 16 weeks post treatment is a longer period of time compared to some of the cited articles. A suggestion is to discuss this (se below).
Outcomes:
Authors equals p16 status to HPV status in table 1 and text, which is not necessarily true, especially since some tumour sites outside the tonsills and base of tongue are included.
Discussion:
Result from other authors (Liu, Vainshtein) with different protocols for follow up PET (12w, 3 months resp) are compared to the 16 weeks protocol. A discussion on how this possibly influences the results is missing.
Indeed it is made very clear that the authors suggest a second look FDG PET for positive or equivocal results at first follow up FDG PET, but for practical clinical purposes it would also be interesting to discuss or comment on which results on the first FDG PET (if any) that should leed to immediate salvage treatment.
Author Response
Reviewer 3
Patient selection:
Data on the 714-88= 626 patients who did not go to a second-look PET are missing. Are all those CMR or are some patients having progressive disease and thus subject to salvage treatment? It should be noted if all patients with positive or equivocal scans at 16 weeks were subject to second-look, and if they were it should be discussed what the purpose of the first scan at 16 weeks is.
Thank you for highlighting this aspect. The “Response Assessment” section of the methods has been expanded to clarify which patients were considered for a second look PET-CT. As you suggest most patients with likely residual disease at initial response assessment were worked up for potential salvage surgery and only a minority of patients in this category underwent second look PET-CT. The outcome of the 626 patients not undergoing second look PET-CT has been added to the results as suggested.
Response assessment:
Response evaluation at 16 weeks post treatment is a longer period of time compared to some of the cited articles. A suggestion is to discuss this (see below).
Our standard practice for a slightly delayed initial response assessment PET-CT is based on previously published work by our group and others. In our experience, we find that this reduces the overall number of equivocal/indeterminate scans compared to imaging at earlier time points but clearly does not eradicate this category as has already been included in the Discussion. The relevant section has been amended to provide more clarity on our rationale for using a delayed initial response assessment PET-CT at 4 months post CRT and reference to the relevant prior paper.
Outcomes:
Authors equals p16 status to HPV status in table 1 and text, which is not necessarily true, especially since some tumour sites outside the tonsils and base of tongue are included.
Thanks for this comment. We concur that p16 expression status isn’t relevant outside of oropharynx sub-sites in HNSCC and HPV status is only prognostic in oropharyngeal SCC. 96% of our patient cohort had tonsillar or base of tongue primary sites and only the remaining 4 patients had rarer oropharyngeal primary sites. There is a very low rate of discrepancy between p16 expression and HPV detected by direct methods. A recent Royal College of Pathologists publication from December 2021 “Dataset for the histopathological reporting of carcinomas of the oropharynx and nasopharynx” states that there is no consensus on the best methodology for HPV testing and that the WHO, AJCC, UICC and a College of American Pathologists Expert Panel all recommend p16 immunohistochemistry to infer HPV status. That said it is acknowledged that where HPV status is used for prognosis and prediction (which is not the case in our work) that HPV-specific testing in addition to p16 expression is used in oropharynx SCC. A comment has been added to the study limitations section of the Discussion regarding this and the above publication has been added as a citation.
Discussion:
Result from other authors (Liu, Vainshtein) with different protocols for follow up PET (12w, 3 months resp) are compared to the 16 weeks protocol. A discussion on how this possibly influences the results is missing.
Thanks, we have expanded this section of the Discussion elaborating on this point and included an additional paper highlighted by Reviewer 1.
Indeed, it is made very clear that the authors suggest a second look FDG PET for positive or equivocal results at first follow up FDG PET, but for practical clinical purposes it would also be interesting to discuss or comment on which results on the first FDG PET (if any) that should lead to immediate salvage treatment.
We have expanded the methods section to clarify our institutional practice on the selection criteria for a second look PET-CT which was previously not clearly communicated. Thank you for highlighting this issue.